# Interaction between Butyrate and Tumor Necrosis Factor α in Primary Rat Colonocytes

**DOI:** 10.3390/biom13020258

**Published:** 2023-01-30

**Authors:** Christopher L. Souders, Juan J. Aristizabal-Henao, Sarah J. Patuel, John A. Bowden, Jasenka Zubcevic, Christopher J. Martyniuk

**Affiliations:** 1Department of Physiological Sciences, College of Veterinary Medicine, University of Florida, Gainesville, FL 32610, USA; 2Center for Environmental and Human Toxicology, University of Florida, Gainesville, FL 32611, USA; 3BERG LLC, 500 Old Connecticut Path, Framingham, MA 01701, USA; 4Department of Chemistry, University of Florida, Gainesville, FL 32611, USA; 5Microbiome Consortium, Center for Hypertension and Precision Medicine, Department of Physiology and Pharmacology, The University of Toledo College of Medicine and Life Sciences, Block Health Science Bldg, 3000 Arlington Ave, Toledo, OH 43614, USA; 6UF Genetics Institute, Interdisciplinary Program in Biomedical Sciences Neuroscience, University of Florida, Gainesville, FL 32611, USA

**Keywords:** hypertension, butyrate, TNFα, colonic epithelium, mitochondria, metabolomics, metabolism, biopterin, ROS1

## Abstract

Butyrate, a short-chain fatty acid, is utilized by the gut epithelium as energy and it improves the gut epithelial barrier. More recently, it has been associated with beneficial effects on immune and cardiovascular homeostasis. Conversely, tumor necrosis factor alpha (TNFα) is a pro-inflammatory and pro-hypertensive cytokine. While butyrate and TNFα are both linked with hypertension, studies have not yet addressed their interaction in the colon. Here, we investigated the capacity of butyrate to modulate a host of effects of TNFα in primary rodent colonic cells in vitro. We measured ATP levels, cell viability, mitochondrial membrane potential (MMP), reactive oxygen species (ROS), mitochondrial oxidative phosphorylation, and glycolytic activity in colonocytes following exposure to either butyrate or TNFα, or both. To address the potential mechanisms, transcripts related to oxidative stress, cell fate, and cell metabolism (Pdk1, Pdk2, Pdk4, Spr, Slc16a1, Slc16a3, Ppargc1a, Cs, Lgr5, Casp3, Tnfr2, Bax, Bcl2, Sod1, Sod2, and Cat) were measured, and untargeted liquid chromatography–tandem mass spectrometry (LC-MS/MS) was employed to profile the metabolic responses of colonocytes following exposure to butyrate and TNFα. We found that both butyrate and TNFα lowered cellular ATP levels towards a quiescent cell energy phenotype, characterized by decreased oxygen consumption and extracellular acidification. Co-treatment with butyrate ameliorated TNFα-induced cytotoxicity and the reduction in cell viability. Butyrate also opposed the TNFα-mediated decrease in MMP and mitochondrial-to-intracellular calcium ratios, suggesting that butyrate may protect colonocytes against TNFα-induced cytotoxicity by decreasing mitochondrial calcium flux. The relative expression levels of pyruvate dehydrogenase kinase 4 (*Pdk4*) were increased via co-treatment of butyrate and TNFα, suggesting the synergistic inhibition of glycolysis. TNFα alone reduced the expression of monocarboxylate transporters slc16a1 and slc16a3, suggesting effects of TNFα on butyrate uptake into colonocytes. Of the 185 metabolites that were detected with LC-MS, the TNFα-induced increase in biopterin produced the only significant change, suggesting an alteration in mitochondrial biogenesis in colonocytes. Considering the reports of elevated colonic TNFα and reduced butyrate metabolism in many conditions, including in hypertension, the present work sheds light on cellular interactions between TNFα and butyrate in colonocytes that may be important in understanding conditions of the colon.

## 1. Introduction

Gut dysbiosis contributes to a variety of conditions including irritable bowel disease, hypertension (HTN), and colon cancer, among others [1,2]. The colonic epithelium is a major site of host–microbiome interactions, and changes in gut microbial composition have profound effects on the host [3,4]. For instance, reduced microbial diversity, and specifically that of butyrate-producing bacterial species, is associated with HTN in rodents and human patients [4,5]. Under physiologic conditions, butyrate is an energy source and a histone deacetylase (HDAC) inhibitor, exhibiting anti-inflammatory, vasoactive [6,7], and anti-hypertensive properties [8,9]. Butyrate modifies mitochondrial metabolism [10] and promotes antioxidant responses [11], a noteworthy point considering the reported alterations in butyrate metabolism in colon cancer [12], colitis [13], atherosclerosis [14], and HTN [5,6]. In view of the latter, we have previously reported a differential butyrate effect on mitochondrial bioenergetics in astrocytes from normotensive compared to hypertensive rodents [15], suggesting alterations in butyrate metabolism in HTN. However, this has not been shown in the colon, where butyrate presents a major source of energy for the colonic epithelial cells.

In contrast, elevated TNFα leads to colonic cell shedding, impairment in colonic barrier integrity [16], and colonic inflammation [17]. TNFα reportedly plays a role in inflammatory bowel disease (Crohn’s disease and ulcerative colitis) [13,18] and in the development of colon cancer [18]. Recently, increased colonic expression of TNFα was reported in the spontaneously hypertensive rat (SHR) [19], a rodent model of HTN characterized by low-grade inflammation, gut dysbiosis, and gut pathology. Moreover, elevated TNFα signaling can alter metabolism in a variety of cell types in favor of anaerobic glycolysis [20,21], in part via reduced oxidative phosphorylation [22]. Importantly, TNFα can modify butyrate’s effects in the colon as it increases glycolysis [21], reduces oxidative phosphorylation [22], and reduces the absorption of butyrate in the GI tract. Reduced butyrate absorption in the GI tract has also recently been documented in the SHR, while elevated fecal butyrate levels and reduced plasma butyrate in human hypertensive patients [23] suggest that reduced butyrate absorption may also be a mechanism in hypertensive patients. Considering the reported opposing effects of butyrate and TNFα in the colon and in HTN, elucidating the cellular interaction of TNFα and butyrate in the colonic epithelium may contribute to our understanding of host–microbiome interactions in the pathophysiology of HTN and other conditions associated with gut dysbiosis.

The current study examined the effects of increasing concentrations of TNFα and butyrate, individually and in co-treatment, on an array of cellular functions in rat primary colonocytes. The results of the current study improve our mechanistic understanding of the effects of TNFα and butyrate in the colon on mitochondrial metabolic endpoints and other cellular functions and shed light on the previously undocumented synergistic effects of butyrate and TNFα on colonic biopterin.

## 2. Materials and Methods

### 2.1. Reagents

The reagents and kits used in the experiments include sodium butyrate (Sigma Aldrich, purity 98%, Cat# 303410), rat TNFα (Sigma Aldrich, purity ≥98%, Cat# T5944), oligomycin A (Sigma Aldrich, purity ≥ 99%, Cat# 75351), 4-trifluoromethoxyphenylhydrazone (FCCP) (Sigma Aldrich, purity ≥ 98%, Cat# C2920), antimycin A (Sigma Aldrich, purity not stated, Cat# A8674), rotenone (Sigma Aldrich, purity ≥95%, Cat# R8875), glucose (Sigma Aldrich, purity ≥ 99.5%, Cat# G8270), and 2-deoxyglucose (Sigma Aldrich, purity ≥ 98%, Cat# D8375), which were all purchased from Sigma Aldrich. The microplate assays included CellTiter-Glo^®^ (Promega, Cat# G9241), AlamarBlue™ (Invitrogen, Cat# A50100), CellTox™ Green (Promega, Cat# G8741), Mitochondrial Membrane Potential Kit (Sigma-Aldrich, Cat# MAK159), and H2DCFDA dye (Invitrogen, Cat# D399). 

The experiments were performed in physiologically normal (Sprague Dawley) rodent colonic epithelial cells rather than human cells, due to inherent ethical issues with collecting physiologically normal colonocytes from humans. Our research focus is also on spontaneously hypertensive rodents as a model, and we aim to study corresponding responses in vivo in rodent models. Doses were selected to cover a broad range of physiological butyrate concentrations and both physiological and pathological colonic TNFα concentrations. Specifically, TNFα has been reported in the range of 50 pg/mL in Sprague Dawley rats and 150–1000 pg/mL in various rodent models of hypertension [24,25]. Butyrate reaches millimolar concentrations luminally but is rapidly metabolized by colonocytes to protect the stem cell niche [26,27], while butyrate concentrations in the portal vein are reported to be in the lower micromolar range.

### 2.2. Cell Culture

Primary rat colonic epithelial cells (Cell Biologics, Cat# RN-6047) were maintained in Complete Epithelial Media (Cell Biologics, M6621), passaged using Hanks Balanced Salt Solution (HBSS) (ThermoFisher, Cat# 14025076) and TrypLE Express (ThermoFisher, Cat# 12604021), and counted used a hemocytometer. 

### 2.3. AlamarBlue™ (Cell Viability) and CellTox™ Green (Cytotoxicity)

Cells were seeded onto a black 96-well plate (BrandTech Scientific, Essex, CT, USA, CAT# 91-425TB) at 10,000 cells per well in 100 µL of medium. The following day, cells were exposed in triplicate (6 wells for medium control) to increasing doses of TNFα (50 pg/mL up to 10 ng/mL), butyrate (50 µM up to 25 mM), as well as co-treatments of 500 µM butyrate and a subset of TNFα doses (50 pg/mL–1 ng/mL) by adding 100 µL of a 2× solution to each well. The lysis control supplied with the CellTox™ Green kit served as the cytotoxic positive control and was added 20 min prior to the measurement of cytotoxicity as a ½ dilution of a 2× stock. On the day of the assay, 100 µL was then removed prior to the addition of 100 µL of CellTox™ Green reagent using a multichannel pipettor. The 96-well plate was mixed via orbital shaker for 30 s at 400 rpm. The plate was then incubated at room temperature for 15 min before fluorescence was measured at an excitation/emission (ex/em) of 485/520 nm on a BioTek Synergy™ 4 Hybrid Microplate Reader using the autogain feature. Following this, 22.2 µL of AlamarBlue™ reagent was added to each well (i.e., a 1/10 dilution) and the plate was incubated at 37 °C for 2 h before an absorbance of 570 nm was measured using the same microplate reader. Wells containing media but no cells were used to remove the average of the background signal. The AlamarBlue™ and CellTox™ Green assay values were normalized to the negative control (containing media only). The data represent one technical replicate (N = 6 wells/technical replicate for medium control and n = 3 wells/technical replicate for experimental conditions).

### 2.4. CellTiter-Glo^®^ Luminescent Cell Viability Assay (Relative ATP Levels)

Cells were seeded onto a white 96-well plate for each technical replicate (BrandTech Scientific, CAT# 91-425TW) at 10,000 cells per well in 100 µL of medium. The following day, cells were exposed in quadruplicate to increasing doses of TNFα (50 pg/mL to 10 ng/mL) and butyrate (50 µM to 25 mM) by adding 100 µL of a 2× solution to each well. As concentrations for TNFα in the literature are typically described as weight per volume (e.g., pg/mL), while butyrate is recorded using molar concentration (e.g., mM), we report both in this format. On the day of the assay, 100 µL was then removed prior to the addition of 100 µL of CellTiter-Glo^®^ 2.0 reagent with a multichannel pipettor. The 96-well plate was mixed via orbital shaker for 30 s at 400 rpm. After 10 min of incubation at room temperature, the luminescent signal was recorded on a BioTek Synergy™ 4 Hybrid Microplate Reader using the autogain feature. The data represent the combination of two technical replicates normalized to the medium control (n = 4 wells/technical replicate for experimental conditions).

### 2.5. Mitochondrial Membrane Potential

Cells were seeded onto a black 96-well plate at 20,000 cells per well in 100 µL of medium. The following day, the plate was exposed in triplicate to increasing doses of TNFα (50 pg/mL up to 5 ng/mL), butyrate (50 µM up to 7.5 mM), as well as co-treatment of 500 µM butyrate and TNFα (50 pg/mL–5 ng/mL) by adding 100 µL of 2× chemical stocks. FCCP is a protonophore that allows hydrogen to pass freely through lipid bilayers and served as a positive control for the loss of mitochondrial membrane potential (MMP). After 24 h of incubation, 100 µL was removed. JC10 dye solution A was prepared by adding 50 µL of JC10 dye to 5 mL of reagent A and vortexing. Next, 50 µL of this mixture was added to each well using a multichannel pipette. The plate was then mixed via an orbital shaker at 400 rpm for 30 s; then, it was incubated in 5% CO_2_ at 37 °C for 1 h, before 50 µL of reagent B was added to each well. The 96-well plate was mixed via orbital shaker for 30 s at 400 rpm. Fluorescence was measured at an ex/em of 540/590 (red) and 490/525 (green) on a BioTek Synergy™ 4 Hybrid Microplate Reader using the autogain feature. The ratio of red/green fluorescence intensities was calculated in Excel (Microsoft 365, software version 2110). Wells containing media but no cells were used to remove the average of the background signal. MMP assay values were normalized to the negative control (containing media only) by dividing everything by the average values for this control. The data represent the combination of two technical replicates normalized to the medium control (n = 3 wells/technical replicate for experimental conditions).

### 2.6. Intracellular and Mitochondrial Calcium

Colon epithelial cells were seeded at 20,000 cells/100 µL/well in a black 96-well plate and allowed to adhere overnight. Cells were then exposed (n = 6 wells/treatment) for a 24 h time period by adding 100 µL of 2× reagent in media to each well. The treatments included media only, positive control, no cell wells, 500 µM, 1000 µM, 2500 µM, and 5000 µM butyrate, or 250 pg/mL, 500 pg/mL, 750 pg/mL, and 1000 pg/mL TNFα plus and minus 500 µM butyrate. The positive control included 5 µM FCCP, as the mitochondrial calcium indicator Rhod-2 is dependent on the mitochondrial membrane potential. A total of 50 µL of medium containing 5 µM Rhod-2 and 20 µM Fluo-5F (i.e., 5× concentrations) was then added to each well using a multichannel pipette. The final concentrations were 1 µM for Rhod-2 and 4 µM for Fluo-5F. Following the exposure and addition of calcium indicators, the 96-well plate was mixed via orbital shaker at 400 rpm for 30 s and returned to the 37 °C CO_2_ incubator. After 30 min of incubation with the calcium indicators, the 96-well plate was put into the machine for a 1 h kinetic run with reads every 5 min spanning the 24 h time period. The reads consisted of excitation/emission values of 552/581 nm for Rhod-2 and 490/525 nm for Fluo-5F, with automatic gain, measured as a bottom read. The background was removed from the data by subtracting the no-cell-fluorescence values from all groups. The data represent one technical replicate normalized to the medium control (n = 6 wells/technical replicate for experimental conditions).

### 2.7. H2DCFDA Assay (ROS Production)

Cells were seeded onto a black 96-well plate (BrandTech Scientific, CAT# 91-425TB) at 25,000 cells per well in 100 µL of medium. The following day, the plate was exposed in triplicate to increasing doses of TNFα (50 pg/mL to 5 ng/mL), butyrate (50 µM to 7.5 mM), as well as co-treatment of 500 µM butyrate and TNFα (50 pg/mL–5 ng/mL) by adding 100 µL of 2× chemical stocks. Next, 50 µM menadione was added 4 h prior to the 24 h time period and served as a positive control for ROS production. After 24 h of incubation, 100 µL was removed. A 40 µM H2DCFDA dye solution (2×) was prepared and vortexed. Next, 100 µL of this mixture was added to each well using a multichannel pipette. The plate was mixed at 400 rpm via orbital shaker; then, it was incubated in 5% CO_2_ at 37 °C for 45 min. Fluorescence was measured at an ex/em of 485/535 nm on a BioTek Synergy™ 4 Hybrid Microplate Reader using the autogain feature. The data represent the combination of two technical replicates normalized to the medium control (n = 3 wells/technical replicate for experimental conditions).

### 2.8. Seahorse Flux Assays (Cell Mitochondrial and Glycolytic Stress Tests)

Cells were seeded onto a cell microplate (Agilent, CAT# 100882-004) at 25,000 cells per well in 100 µL of medium. To prevent edge effects following seeding, cells were left at room temperature for 30 min prior to incubation at 37 °C. Following cell attachment in 6–8 h, 150 µL was added. The following day, the plate was exposed by adding an additional 250 µL of 2× chemical stocks. Following 24 h exposure, the cells were washed three times by removing, and then, adding 350 µL of warm Krebs Heinseleit buffer +200 µM glucose lacking sodium bicarbonate (pH 7.1 to 7.2 before sterile filtering). Following washes, the cells were transferred to a 37 °C non-CO_2_ incubator for 1 h while the flux plate (Agilent, CAT# 102340-100) was loaded with 10× injection chemicals. The Glycolytic Stress Test and Cell Mitochondrial Stress Test consisted of a constant concentration a 10× injection scheme. The Glycolytic Stress Test consisted of the sequential injection of 56 µL of 250 mM glucose, followed by 62 µL of 20 µM oligomycin, and finally, 69 µL of 1M of 2-deoxyglucose. The Cell Mitochondrial Stress Test consisted of the sequential injection of 10× chemicals, specifically 56 µL of 7.5 µM oligomycin, followed by 62 µL of 30 µM FCCP, and finally, 69 µL of 10 µM of antimycin A and rotenone. 

### 2.9. Gene Expression Analysis

Five separate flasks of cells were treated with either butyrate (500 µM), TNFα (500 pg/mL), or co-treatment together (500 µM butyrate and 500 pg/mL TNFα) for 24 h as per the methods outlined above. The 500 pg/mL TNFα is within the range of serum concentrations for different hypertensive models, including the SHR (150 to 300 pg/mL) and the Sabra rat model of salt-sensitive hypertension (150 to 1100 pg/mL) [24,25], whereas 500 µM butyrate is below cytotoxicity and has been shown to improve inflammatory signaling in colonocytes [28]. Real-time PCR analysis followed our established methods [29]. Colon epithelial cells were briefly homogenized in TRIzol Reagent (Life Technologies, Carlsbad, CA, USA) and RNA extracted as per the manufacturer’s protocol. A 2100 Bioanalyzer (Agilent) was used to evaluate RNA quality (RNA integrity values > 7). The cDNA synthesis proceeded with 750 ng RNA using the iScript (BioRad) kit followed by DNase treatment (Turbo DNase, Ambion) as per the manufacturer’s instructions. No reverse transcriptase (NRT) controls (n = 3) and 2 no-template controls were prepared and included in all qPCR reactions. NRTs were constructed from a pool of 3 different RNA samples chosen at random, and this was carried out 3 times. The cDNA synthesis reaction proceeded as follows: 5 min at 25 °C, 30 min at 42 °C, and 5 min at 85 °C using a MyCycler™ Thermal Cycler. The reaction mixture was then diluted at 1:20 with nuclease-free water and used for real-time PCR amplification. 

Target genes were selected based upon their role in mitochondrial oxidative respiration, glycolysis, cell differentiation, oxidative stress, and apoptosis. These genes included Pdk1, Pdk2, Pdk4, Spr, Slc16a1, Slc16a3, Lgr5, Casp3, Tnfr2, Bax, Bcl2, Cs, Ppargc1a, Sod1, Sod2, and Cat. Appendix A contains all the sequence information for target genes. The reference and target genes were measured using the CFX Connect Real-Time PCR Detection System (BioRad) using SsoFast™ EvaGreen^®^ Supermix (BioRad, Hercules, CA, USA). Biological replicates (n = 5 per experimental group) were measured in duplicate. Two reference genes (ribosomal subunit 18 (rps18) and beta actin (β-actin)) were used to normalize target expression. The M-value and coefficient of variation for the two combined reference genes were M = 1.34 (CV = 0.46) for cDNA synthesis 1 and M = 0.66 (CV = 0.23) for cDNA synthesis 2. Due to the number of transcripts assessed, two cDNA synthesis reactions were conducted. In each case, genes were normalized to rps18 and β-actin. Normalized expression based upon the relative ΔΔCq method was obtained for each target gene using CFX Manager™ software (v3.1) (baseline subtracted) [30]. Real-time PCR data followed recommendations outlined in the MIQE guidelines [31].

### 2.10. Cell Culture Exposure for Metabolomics

Colon epithelial cells were exposed in T75 flasks for 24 h to either medium, 500 µM butyrate, 500 pg/mL TNFα, or a combination of butyrate and TNFα (N = 8/group) by removing the media and replacing them with 8 mL of 1× chemical stocks. Following this exposure, cells were passaged in groups of 4 (containing one individual from each treatment group to minimize exposure length differences) by removing the media and washing the flask with 10 mL of sterile Hank’s Balanced Salt Solution (HBSS), then with 3 mL of TrypLE Express for 2 min in a 37 °C CO_2_ incubator, before the cells were transferred to a 15 mL conical flask with 4× the volume of HBSS. The groups of four were then centrifuged for 5 min at 300× *g*. The cell pellet was then aseptically resuspended in 1 mL of ice-cold rinsing solvents (0.3% ammonium acetate, 0.3% ammonium formate, 0.9% NaCl, 1 M PBS, and 100 mM HEPES), transferred to a pre-weighed and labeled 1.7 mL microtube (Olympus, CAT# 22-281), and centrifuged at 300× *g* for 5 min. The supernatant was then discarded and the cells were transferred to a −80 °C freezer until extraction.

### 2.11. Metabolomics Extraction and Analysis via Ultra-High-Performance Liquid Chromatography–Tandem Mass Spectrometry

Medium-free cell pellets (approximately 50 mg/sample) or quality control samples (NIST SRM 1950, 100 µL per sample) were thawed on ice. Known amounts of isotopically labeled internal standards in water with 0.1% formic acid (Metabolomics QC Kit, Cambridge Isotope Laboratories, Tewksbury, MA, USA) were added to each sample, followed by the addition of 1200 µL ice-cold (−20 °C) 8:1:1 acetonitrile:methanol:acetone (*v*/*v*/*v*; Optima grade) and vigorous vortexing for 30 min. The samples were centrifuged at 15,000× *g* at 4 °C for 30 min. The supernatants were carefully extracted, transferred into new tubes, and evaporated fully under a stream of nitrogen gas at 30 °C. Dried metabolite extracts were reconstituted in 100 µL 95:2:2:1 water:methanol:acetonitrile:formic acid (*v*/*v*/*v*/*v*) and were saved at 4 °C until analysis.

Metabolomic analyses were completed using a Thermo Vanquish UHPLC system coupled to a Thermo Q-Exactive Orbitrap mass spectrometer. A Waters Acquity HSS-T3 column was used (150 mm × 2.1 mm × 1.8 µm), and the mobile phase consisted of (A) 100% water and (B) 100% acetonitrile, both with 0.1% formic acid. The multi-step gradient was as follows: Starting at 1% B, hold until the 3 min mark; then, linearly increase to 50% B between 3 and 11 min, followed by a linear increase to 98% B between 11 and 12 min. Hold at 98% B between 12 and 16 min, decrease to 1% B at 16.1 min, and hold at 1% B until the 20 min mark. The column compartment was kept at 45 °C and the flow rate was 0.25 mL/minute. The injection volume was 10 µL and the autosampler temperature was 4 °C. The mass spectrometer was operated in full-scan positive and negative ion modes (two separate injections per sample), with spray voltages of +3.5 kV and −3.5 kV, respectively. The scan range was *m*/*z* 50–750, the resolution was 70,000, the sheath gas flow rate was 40, the aux gas was 8, the capillary temperature was 275 °C, and the aux gas heater temperature was 320 °C. Data-dependent tandem mass spectrometry experiments (ddMSMS) were performed using pooled samples created by mixing equal parts of extracts from each sample for each experimental condition. IE-Omics [32] was used to generate iterative exclusion lists in three rounds of top-10 ddMSMS in both positive- and negative-ion modes (6 injections per pool). In the ddMSMS experiments, the resolution was 35,000 in full-scan and 17,500 in MSMS. The isolation window was 1 m/z, the stepped normalized collision energy was 20, 25, and 30, and dynamic exclusion was set at 6 s. Three extraction blanks (containing internal standards) and solvent blanks (without internal standards) were also included in the analysis. All the samples, quality controls, blanks, and pools were randomized within the batch sequence.

Metabolite identifications were completed using ThermoFisher Compound Discoverer software (v.3.2), using the following settings: precursor mass tolerance: 5 ppm, maximum retention time shift: 30 s, mzCloud match factor threshold: 95%. Compounds without MSMS (identified only using accurate mass) and those without a full mzCloud match were filtered out. Redundant features were collapsed by selecting the single identification with the highest mzCloud match score (including repeated identifications within/between runs and between positive and negative ionization modes). Redundant features with the same mzCloud score were collapsed by selecting the single feature with the highest peak area. Peak areas were normalized using isotopically labeled (13C6) phenylalanine in both positive- and negative-ion modes (*m*/*z* 172.10692 and 170.09126, respectively). Semiquantitative concentrations are expressed as µg metabolite per g cells (µg/g, “SemiQuan_CONC” sheet in Appendix A) for 185 metabolites. There appear to be two outliers, as shown in the “PCAscores_ALL” tab, and both are in the TNFa + Butyrate group. The quality control samples (SRM 1950) were also included within the analysis.

### 2.12. Statistical Analysis

Statistical analyses were conducted in GraphPad Prism (version 9.0). The data were tested for normality using the Shapiro–Wilk test and for homoscedasticity using the Brown–Forsythe test. the Relative ATP levels, cell viability, cytotoxicity, ROS, and MMP data were analyzed using a Kruskal–Wallis test followed by a Dunn’s multiple comparison test to determine significance. To simplify comparisons and increase statistical power, comparisons on graph depict those to the medium control only. However, in Appendix A, all group comparisons across treatment concentrations are presented. The log transform of mitochondrial to intracellular calcium satisfied both normality and homoscedasticity and was therefore analyzed using a One-way ANOVA followed by a Dunnett’s multiple comparisons test. Gene expression data were analyzed using a One-way ANOVA followed by a Dunnett’s multiple comparisons test. MST and GST graphs were calculated following the Seahorse XF Cell Mito Stress Test Kit User Guide (User Guide Kit 103015−100, Agilent) and analyzed using a One-Way ANOVA followed by a Dunnett’s multiple comparisons test after the data were checked for normality.

Differences in the metabolomic profiles between groups were assessed via principal component analysis (PCA) using log-transformed, Pareto-scaled (mean-centered divided by the square root of the standard deviation of each variable), and internal standard-normalized semi-quantitative peak areas using MetaboAnalyst 5.0 software [33]. Differences in the amounts of individual metabolites between groups were assessed via One-way ANOVA with Fisher’s LSD post hoc test (statistical significance was inferred at *p* < 0.05 (false discovery rate (FDR)-adjusted)). Volcano plots were created comparing selected groups; highlighted metabolites in the upper left and right quadrants represent statistically significant features with FDR-adjusted *p*-values < 0.05 and a fold change threshold of 2 (log2-fold change < −1 or > 1). For this experiment, the quality control SRM1950 technical triplicates were clustered tightly within the PCA scores plot and the median percent coefficient of variation for these samples for all metabolites was 13.2%, and 9.59% for biopterin (“PCA scores plot” sheet in Appendix A).

## 3. Results

### 3.1. Cell Viability and Cytotoxicity

We first tested the effects of increasing concentrations of butyrate and TNFα on cell viability and cytotoxicity in rat primary CECs. Considering the wide range of physiologic concentrations reported in the literature, we aimed to establish the temporal (4, 24, and 72 h treatments) and dose-dependent interaction between the two agents in rat CEC culture. Based upon these data, we chose to investigate only the 24 h time point for mitochondrial bioenergetics, gene expression analysis, and metabolomics.

The experimental groups varied for cell viability (Kruskal–Wallis test statistic = 89.55, number of groups = 29, *p* < 0.0001) and cytotoxicity (Kruskal–Wallis test statistic = 88.02, number of groups = 29, *p* < 0.0001) in primary rat CECs with butyrate and TNFα concentrations at 24 h (Figure 1A,B). A decrease in cell viability was detected between the medium control and concentrations of > 1 ng/mL TNFα (*p* < 0.05) (Figure 1A). Butyrate at 5 mM and above reduced viability compared to media alone (*p* < 0.05). Co-treatment of 500 µM butyrate over increasing concentrations of TNFα (500 pg/mL–1 ng/mL) did not affect cell viability (Figure 1A). When CECs were cotreated with 500 µM butyrate and 1 ng/mL TNFα, butyrate co-treatment had a protective effect on cell viability compared to 1 ng/mL TNFα alone. These comparisons are based upon treatment comparisons to the medium control. 

TNFα treatment alone increased CEC cytotoxicity (by 125–600%, Figure 1B), at concentrations > 1 ng/mL TNFα relative to untreated cells. Butyrate treatment alone (up to 25 mM) did not induce cytotoxicity compared to untreated cells (Figure 1B). Co-treatment of 500 µM butyrate with 1 ng/mL TNFα, however, appeared to have a protective effect on TNFα-induced cytotoxicity when compared to 1 ng/mL TNFα alone (for statistical comparisons made amongst all groups, see Appendix A). These data suggest dose-specific protective effects of butyrate on TNFα-induced reductions in cell viability and cell cytotoxicity in rodent CECs. 

### 3.2. Relative ATP Levels

Considering a major role for butyrate as an energy source for CECs, we next investigated the interaction between butyrate and TNFα in ATP production in rat CECs. The experimental groups for relative ATP levels varied in primary rat CECs with butyrate and TNFα treatment at 4 h (Kruskal–Wallis test statistic = 141.7, number of groups = 30, *p* < 0.0001), 24 h (Kruskal–Wallis test statistic = 157.8, number of groups = 33, *p* < 0.0001), and 72 h (Kruskal–Wallis test statistic = 132.7, number of groups = 23, *p* < 0.0001) (Figure 2, Appendix A). At 4 h, ATP levels had not changed with TNFα treatment alone; however, treatment with butyrate at different concentrations lowered ATP levels at 4 h (Appendix A). This was also observed following co-treatment with TNFα and 500 µM butyrate at 4 h (*p* < 0.05, Appendix A). At 24 h (Figure 2), treatment with TNFα alone decreased ATP levels relative to the medium control, at concentrations of >1 ng/mL. At higher concentrations (>2.5 mM), butyrate treatment alone also decreased ATP levels relative to the medium control (*p* < 0.05, Figure 2). Co-treatment with TNFα and 500 µM butyrate reduced ATP levels at higher concentrations of TNFα (>2.5 ng/mL, *p* < 0.05 compared to the medium control, Figure 2). An intergroup comparison showed no significant protective effects of butyrate on TNFα-induced ATP reduction (for statistical comparisons made amongst all groups, see Appendix A). At 72 h, ATP levels were reduced by 250 pg/mL TNFα alone, and in combination with 25 µM and 250 µM butyrate relative to the control (*p* < 0.05) (Appendix A). These data suggest no major interactions between butyrate and TNFα regarding ATP production in rodent CECs.

### 3.3. Mitochondrial Membrane Potential

We next assessed interactions between butyrate and TNFα as it pertains to mitochondrial function by assessing their effects on MMP. The experimental groups differed for MMP in primary rat CECs with butyrate and TNFα concentration at 4 h (Kruskal–Wallis test statistic = 61.88, number of groups = 30, *p* = 0.0004), 24 h (Kruskal–Wallis test statistic = 151.9, number of groups = 30, *p* < 0.0001), and 72 h (Kruskal–Wallis test statistic = 89.88, number of groups = 23, *p* < 0.0001) (Figure 3, Appendix A). At 4 h, there was no change in MMP with any treatment (Appendix A). At 24 h, 200 and 5000 pg/mL TNFα decreased MMP by approximately 50%, while there was a general increase (by 45–130%, Figure 3) in MMP with 75 µM to 5 mM butyrate treatment. Co-treatment with 500 µM butyrate and 50 to 500 pg/mL TNFα increased MMP relative to the medium control (Figure 3). A different trend was observed at 72 h, and there was little change compared to the untreated cells. The only change was observed with 250 µM butyrate, which increased MMP (*p* < 0.05) (Appendix A). Taken together, TNFα and butyrate exerted different effects on MMP at 24 h relative to the other time points investigated, more so than other time points. There was no significant interaction effect observed between butyrate and TNFα on MMP (for statistical comparisons made amongst all groups, see Appendix A).

### 3.4. Mitochondrial and Intracellular Calcium

Calcium is important for stabilizing the membranes of mitochondria and signal transduction. Calcium levels in primary rat CECs differed among groups with butyrate and TNFα at 24 h for mitochondrial calcium (F_(13,73)_ = 55.78, *p* < 0.0001) (Appendix A), intracellular calcium (F_(13,73)_ = 10.68, *p* < 0.0001) (Appendix A), as well as the mitochondrial-to-intracellular calcium ratio (F_(13,73)_ = 91.06, *p* < 0.0001) (Figure 4). At 24 h, mitochondrial calcium was increased by 250 and 500 pg/mL TNFα (a 30% and 40% increase relative to untreated cells, respectively) (Appendix A) compared to untreated cells. Conversely, intracellular (or cytosolic) calcium was significantly decreased by TNFα treatment alone (at 500–1000 pg/mL, Appendix A) compared to untreated cells. Butyrate treatment alone did not affect mitochondrial or cytosolic calcium at any concentration compared to untreated cells (Appendix A). However, when plotting the ratio between mitochondrial and cytosolic calcium (Figure 4), it was noted that TNFα treatment increased the mitochondrial/intracellular calcium ratio, while butyrate treatment decreased the ratio at lower concentrations of butyrate. The ratio is important because changes in calcium balance can be related to changes in the ATP production and metabolic capacity of CECs. Co-treatment of TNFα with butyrate did not significantly alter the calcium ratios compared to untreated cells, suggesting that butyrate may potentially stabilize the overall CEC calcium levels that are perturbed by TNFα.

### 3.5. ROS Production

TNFα and butyrate are known immune modulators, and ROS is a known hallmark of inflammatory disorders including HTN, but the interaction between TNFα and butyrate in the production of ROS in rat CECs, to our knowledge, has not been investigated to date. At 4 h, there was no ROS induction detected with any treatment (Appendix A) (Kruskal–Wallis test statistic = 148.6, number of groups = 30, *p* < 0.0001). At 24 h, TNFα alone produced a significant 30% and 90% decrease in ROS at the two highest concentrations only (2500 and 5000 pg/mL, respectively, Figure 5) when compared to untreated cells. Butyrate treatment alone did not affect ROS levels (Figure 5). Co-treatment with butyrate and TNFα decreased ROS when compared to untreated cells, with a 55–100% decrease observed at several concentrations of TNFα (200–5000 pg/mL) (Kruskal–Wallis test statistic = 148.4, number of groups = 30, *p* < 0.0001) (Figure 5). Intergroup comparisons, however, showed no significant difference between TNFα treatments alone and those in the presence of butyrate (for statistical comparisons made amongst all groups, see Appendix A), suggesting no significant interaction between TNFα and butyrate in ROS production in rodent CECs. 

### 3.6. Cell Mitochondrial Stress Test

For an in-depth analysis of mitochondrial function and considering the role of butyrate in mitochondrial respiration, we investigated the interactions between butyrate and TNFα in mitochondrial respiration using a Mitochondrial Stress Test (Figure 6A). First, we investigated the effects of increasing concentrations of TNFα. Mitochondrial respiration in primary rat CECs varied following treatment with different TNFα concentrations at 24 h for the energy map (F_(4,14)_ = 5.52, *p* = 0.0070) (Figure 6B), basal respiration (F_(4,14)_ = 6.21, *p* = 0.0043) (Figure 6C), ATP-linked respiration (F_(4,14)_ = 6.10, *p* = 0.0047) (Figure 6D), maximal respiration (F_(4,14)_ = 9.19, *p* = 0.0007) (Figure 6E), spare capacity (F_(4,14)_ = 8.37, *p* = 0.0012) (Figure 6F), and non-mitochondrial respiration (F_(4,14)_ = 2.90, *p* = 0.0609) (Figure 6H), but not proton leak (F_(4,14)_ = 3.45, *p* = 0.0369) (Figure 6G), compared to untreated cells. Both 250 and 500 pg/mL TNFα treatment resulted in energy maps with a decrease in both extracellular acidification and oxygen consumption, indicating a more quiescent phenotype (Figure 6B). 

We next investigated whether butyrate could modify the effects of select concentrations of TNFα on mitochondrial respiration in rodent CECs. Mitochondrial respiration in primary rat CECs varied with butyrate, TNFα, and co-treatment at 24 h for the energy map (F_(5,13)_ = 10.85, *p* = 0.0003), basal respiration (F_(5,13)_ = 16.93, *p* < 0.0001), ATP-linked respiration (F_(5,13)_ = 30.25, *p* < 0.0001), and maximal respiration (F_(5,13)_ = 5.10, *p* = 0.0083) (Figure 7C–E) but did not differ significantly for spare capacity (F_(5,13)_ = 1.78, *p* = 0.1861), proton leak (F_(5,12)_ = 1.90, *p* = 0.1678), nor non-mitochondrial respiration (F_(5,13)_ = 1.86, *p* = 0.1712) (Figure 7F–H). Interestingly, butyrate treatment alone significantly lowered basal and ATP-linked, but not maximal respiration, when compared to untreated cells (Figure 7C–E). Butyrate was, however, unable to affect the TNFα-dependent effects on these variables (Figure 7C,D), suggesting no interactions between butyrate and TNFα on mitochondrial respiration in CECs. Spare capacity, proton leak, and non-mitochondrial respiration were not significantly different across all groups (Figure 7F–H).

### 3.7. Glycolytic Stress Test

We next tested the effect of increasing concentrations of TNFα of glycolytic metabolism in rodent CECs. Glucose is the primary energy source for CECs and a shift in energy preference can indicate a change in metabolic requirements or cellular stress. Following 24 h treatment, glycolytic respiration in primary rat CECs varied with TNFα concentration for the energy map (F_(4,15)_ = 6.10, *p* = 0.0041), glycolytic reserve (F_(4,15)_ = 10.32, *p* = 0.0003), and non-glycolytic acidification (F_(4,15)_ = 4.36, *p* = 0.0155), but not glycolysis (F_(4,15)_ = 1.83, *p* = 0.1762) or glycolytic capacity (F_(4,15)_ = 6.211, *p* = 0.0037) when compared to the control group (Figure 8A–F). Again, the energy maps revealed a decrease in both extracellular acidification and oxygen consumption, indicating a more quiescent phenotype (Figure 8B). Both 250 pg/mL and 500 pg/mL TNFα lowered the glycolytic reserve by ~60 and 85%, respectively (Figure 8E), when compared to untreated cells. In addition, 250 pg/mL and 500 pg/mL TNFα significantly reduced non-glycolytic acidification (Figure 8F) when compared to untreated cells. TNFα did not significantly alter glycolysis or glycolytic capacity at the doses tested relative to control (Figure 8C,D). 

Next, we tested whether co-treatment with butyrate could modify the TNFα-dependent effects of glycolytic metabolism in rodent CECs. Glycolytic respiration in primary rat CECs differed with butyrate and TNFα concentration after 24 h, both individually and in co-treatment, for the energy map (F_(5,14)_ = 2.912, *p* = 0.0524), glycolytic acidification (F_(5,14)_ = 2.676, *p* = 0.0671), glycolytic capacity (F_(5,14)_ = 3.736, *p* = 0.0234), glycolytic reserve (F_(5,14)_ = 5.168, *p* = 0.0068), and non-glycolytic acidification (F_(5,14)_ = 22.79, *p* < 0.0001) (Figure 9A–F). As shown in Figure 8, TNFα treatment alone did not change glycolysis or glycolytic capacity (Figure 9A,E) and it reduced both glycolytic acidification and non-glycolytic reserve (Figure 9D,F) when compared to untreated cells. Butyrate alone significantly reduced non-glycolytic acidification only (Figure 9F) when compared to untreated cells. 

Based on our mitochondrial and cytotoxicity/cell viability data, we selected 500 µM butyrate and 500 pg/mL TNFα alone and in co-treatment to further probe mechanisms via gene expression and metabolomics. This concentration of TNFα is within the physiological range of that reported for hypertensive models [24,25], while butyrate has been reported to modulate the downstream signaling of TNFα in colonocytes at 500 µM [28]. Butyrate is reportedly present in the colon at millimolar concentrations luminally but is believed to exist in a gradient within the colonic epithelium, with low levels reaching the stem cell niche due to its high metabolism and mucus secretion as a barrier [26,27]. The selected 500 µM butyrate concentrations therefore represented an intermediate physiologic dose that is present within the colonic epithelium.

### 3.8. Gene Expression

For transcripts related to apoptosis, cell differentiation, and metabolism, several transcripts differed in their relative expression compared to the housekeeping gene and among experimental groups (Figure 10). Compared to the medium control, expression levels of Casp3 (F_(3,16)_ = 6.715, *p* = 0.0038) and Tnfr2 (F_(3,16)_ = 9.044, *p* = 0.0010) were both decreased in cells treated with TNFα in the presence or absence of butyrate. Compared to the medium control, the transcript levels of the epithelial stem cell marker Lgr5 significantly decreased following TNFα treatment alone (F_(3,16)_ = 5.148, *p* = 0.0111) but not in any other treatment group. Compared to the medium control, Pdk4 expression levels were upregulated by TNFα and butyrate co-treatment (F_(3,16)_ = 16.77, *p* < 0.0001) with no changes observed in any other treatment group. The monocarboxylate transporters of the short-chain fatty acids Slc16a1 (F_(3,16)_ = 11.37, *p* = 0.0003) and Slc16a3 (F_(3,16)_ = 2.332, *p* = 0.1128) were downregulated by TNFα treatment, with Slc16a1 expression levels also downregulated by butyrate treatment alone. No significant differences were detected among the groups for transcript levels of Bax, Bcl2, Cat, Sod1, Sod2, Cs, Pdk1, Pdk2, Ppargc1a, and Spr (Figure 10A–P).

### 3.9. Metabolomics

We detected a total of 185 metabolites in the colonocytes with high confidence, across one or more treatments (Appendix A). The PCA analysis revealed that metabolite abundance was relatively consistent between treatments, showing significant overlap, and only one compound, biopterin, was significantly different across groups following ANOVA and an FDR correction (Figure 11A,B). Compared to untreated cells, we observed a significant increase in biopterin levels in the primary rat CECs following TNFα treatment in the presence or absence of butyrate (F_(3,27)_ = 5.640, *p* = 0.0039) (Figure 11C). Butyrate alone did not change biopterin levels in rat CECs. These data suggest no interaction between butyrate and TNFα in the production of biopterin in rodent CECs.

Prior to FDR correction, there were several other metabolites that differed among groups (*p* < 0.05) (Appendix A), which may warrant further investigation in vivo. Noteworthy is the observation that changes in adenosine 5’-monophosphate (AMP) correspond to changes in ATP following TNFα treatment (Figure 2).

## 4. Discussion

This study demonstrated the following novel findings: (i) A significant interaction between butyrate and TNFα was observed at some concentration for cell viability, cell cytotoxicity, and calcium ratios were tested in primary rodent CECs, suggesting that butyrate may protect against TNFα-induced changes in these variables. (ii) No significant interactions between butyrate and TNFα were observed when measuring ATP production, MMP, or ROS. However, one or both treatments alone were able to affect said variables, as discussed in detail in the below text. (iii) TNFα significantly decreased several metrics of CEC respiration and energy production, including glycolytic, reserves and non-glycolytic acidification. Butyrate co-treatment did not modify the TNFα-induced effects on these variables in CECs. (iv) Co-treatment of butyrate and TNFα increased Pdk4 expression but decreased glycolysis, OxPhos, and ATP levels, suggesting effects on epithelial lipogenesis. (v) TNFα-induced biopterin levels in colonocytes were augmented, while co-treatment with butyrate was unable to mitigate the effect on biopterin levels. Figure 12 highlights the proposed interactions between butyrate and TNFα in rodent CECs.

### 4.1. TNFα Impairs Glycolysis and Mitochondrial Respiration despite the Presence of Physiologic Butyrate Levels in Rodent Colonic Epithelial Cells

Our data indicate that both butyrate and TNFα can induce a quiescent energy phenotype in CECs by lowering ATP levels in a dose-dependent manner. Our gene expression data support this, as we observed increased levels of *Pdk4* transcripts during TNFα and butyrate co-treatment. Additionally, as early as 3 weeks of age, the SHRs have significantly elevated expression of the catalytic subunit of AMPK (*Prkaa2*), suggesting a low AMP-to-ATP ratio [34,35]. However, as the physiologic environment of CECs in the colon contains microbial metabolites in vivo, the data obtained in cultured CECs in the presence of butyrate may be more indicative of metabolic homeostasis in vivo. In the presence of physiologic levels of butyrate, TNFα lowered ATP-linked mitochondrial respiration and inhibited glycolysis, glycolytic reserve, and glycolytic capacity. Of note, both individual treatments and the co-treatment of butyrate with TNFα significantly reduced non-glycolytic acidification. Taken together, these data suggest greater ATP-consuming anabolic metabolism in CECs in the presence of TNFα, which may not be counteracted by butyrate. Thus, alterations in mitochondrial function and cell metabolism may exist in conditions of elevated inflammation mediated by TNFα. In support of this, inflammatory bowel disease is linked with decreased mitochondrial metabolism, including lower ATP production [36]. Considering the association between elevated TNFα and HTN [25,37], the deregulated metabolism in rodent models of HTN [38], as well as the high prevalence of metabolic syndrome in hypertensive patients [39], the role of colonic TNFα in HTN warrants further elucidation in vivo.

### 4.2. Butyrate Improves Cell Viability and Decreases TNFα-Induced CEC Cytotoxicity

We first aimed to establish cytotoxic thresholds in primary rat CECs. Our data indicate that micromolar concentrations of butyrate may counteract the cytotoxicity induced by TNFα. Cytotoxicity can be defined as a reduction in metabolic activity with a corresponding decrease in cell membrane potential, whereas cytostatic effects can be described as reductions in metabolic activity without a corresponding decrease in cell membrane potential [40]. Our data suggest that TNFα and butyrate may each be cytostatic at given concentrations. Co-treatment of 500 µM butyrate with 1 ng/mL TNFα appeared to have a protective effect on cytotoxicity when compared to 1 ng/mL TNFα alone. The levels of TNFα in hypertensive colons reportedly fall within cytotoxic levels when considering local and circulating pools of TNFα. In the three-week-old SHRs, colonic TNFα mRNA expression is two times higher than in the age-matched WKY controls [34]. Circulating levels of TNFα have been reported in the range of 150 pg/mL to 300 pg/mL in the SHR [24], and 150 pg/mL to 1100 pg/mL in the Sabra salt-sensitive model of hypertension [25]. Here, we show that co-treatment with 500 µM butyrate can shift the CECs treated with 150 pg/mL TNFα from a cytotoxic to a cytostatic state, indicating that butyrate may exert a protective effect over TNFα in CECs. Co-treatment with butyrate also lowered the cytotoxicity of TNFα at 1 ng/mL concentrations, while improving viability at 1 ng/mL relative to TNFα alone. These data suggest that the interplay between TNFα and butyrate may influence epithelial cell viability in the colon in hypertension and other conditions involving gut dysbiosis and TNFα-mediated inflammation. Considering that butyrate transport across the colonic epithelium is reduced in the SHR [9], TNFα may exert a more mitotoxic phenotype, potentially exacerbating colonic inflammation and dysbiosis in HTN. 

### 4.3. Butyrate Protects against TNFα-Induced Reduction in Mitochondrial Membrane by Stabilizing Mitochondrial–Cytosolic Calcium Levels

Butyrate alone was observed to elevate mitochondrial membrane potential (MMP) in a dose-dependent manner, while TNFα treatment alone decreased MMP. Moreover, the presence of butyrate in the cell culture counteracted the TNFα-induced decrease in MMP, suggesting that butyrate can exert protective effects on cell viability by stabilizing MMP. MMP is the electrical potential (ΔΨm) established by the charge separation of protons in the intermembrane space to generate ATP, to import cations such as calcium and iron, and to facilitate the mitophagy of depolarized mitochondria [41,42]. MMP stability is crucial for overall cell homeostasis [41], and the accumulation of calcium in the mitochondria is known to transiently depolarize MMP; conversely, decreased mitochondrial calcium can downregulate mitochondrial metabolism [43]. Our results indicate that butyrate lowers, while TNFα increases, calcium ratios in the mitochondria following a 24 h treatment. TNFα can reportedly increase the cellular influx of calcium, which can lead to higher ROS production [43,44]. In line with this, TNFα reduced MMP in murine intestinal biopsies in vitro [45], which was prevented by mitoquinone, an antioxidant that accumulates in the mitochondria. Another study reported alterations in MMP in TNFα-sensitive L2929 cells, but not in the TNFα-resistant subclone L2929.12 cell line, which had constitutively higher SOD activity [46]; this is further evidence that TNFα alters MMP directly. 

The detrimental effects of TNFα on MMP may be secondary to increased calcium signaling and ROS production [45,46]. Since ROS production is dependent on MMP [41], the acute effects of butyrate may increase ROS generation by the colonic cells through the hyperpolarization of mitochondria. This short-term increase in ROS may be a protective action by the host against a potential infection threat coming from the gut lumen, which the host perceives via elevated bacterial metabolites and elevated inflammatory cytokines such as TNFα [47,48]. Short-term butyrate exposure has been shown to generate ROS in colonocytes, ostensibly via mitochondrial metabolism rather than receptor signaling, leading to compensatory responses that inhibit NF-κB signaling following TNFα treatment [49]. 

It is noteworthy that at 24 h, butyrate reduced ROS production with or without TNFα, even in the presence of elevated MMP, which is likely related to the quiescent energy phenotype. Pro-oxidants are causally linked to, whereas antioxidants are protective against, HTN [50], suggesting that colonic butyrate may play a complex role in conditions such as HTN. As mentioned above, our findings also suggest that TNFα and butyrate may influence cardiovascular homeostasis by modulating colonic calcium uptake, which, in turn, may also modulate blood pressure [51,52]. Butyrate can modulate calcium uptake via multiple signaling pathways, including via Vitamin D receptor [53] and peroxisomal proliferator-activated receptor γ (PPARγ) [54], whereas TNFα appears to oppose both Vitamin D and PPARγ activity [18,55]. Our group has previously shown that alterations in colonic and brain TNFα and PPARγ signaling were associated with shifts in the gut microbiota and blood pressure alterations in a mouse model associated with reduced circulating and colonic macrophages [56,57,58]. Considering that activated macrophages are a major source of TNFα [59], future investigations should assess butyrate–TNFα host–microbiota relationships in HTN in vivo. 

Although we did not investigate them directly, alterations in MMP are also associated with potential changes in iron uptake in the colon. Specifically, the uptake of iron from the cytosolic labile iron pool (CLIP) to mitochondria has been shown to be dependent on MMP [60,61]. Mitochondria, in turn, play a central role in the production of heme iron, and in vertebrates, most iron is used in the production of hemoglobin in red blood cells [61]. Low iron intake has been correlated with HTN [62] and altered iron metabolism may play a role in metabolic syndrome [63,64]. Therefore, alterations to MMP have important implications for iron homeostasis, suggesting that butyrate may have a role in facilitating mitochondrial iron trafficking, especially when TNFα is also present. 

### 4.4. Butyrate and TNFα Alter Transcripts Involved in Metabolism and Differentiation of CECs

Physiological concentrations of TNFα exerted an inhibitory effect on Tnfr2 and Casp3 expression at 24 h, both in the presence and absence of butyrate. These data suggest pro-survival action, as the inhibition of caspase-3 has been shown to decrease responsiveness to TNFα-induced apoptosis [65]. Additionally, the knockout of Tnfr2 has been shown to increase colonic inflammation by increasing CD8+ T-cells, suggesting that this might be a proinflammatory phenotype [66]. Others have shown that 50 ng/mL TNFα elevated Tnfr2 protein levels in COLO205 and DLD-1 cells following 10 h exposure, but only when co-treated with other inflammatory cytokines [67]. Together, these suggest that, depending on the levels of TNFα, downstream signaling pathways may differ. 

Leucine-rich repeat-containing G protein-coupled receptor 5 (Lgr5) is a stem cell marker essential to colonic homeostasis [68]. Lgr5 expression was reduced in CECs treated with 500 pg/mL TNFα, both in the presence and absence of butyrate. Although we did not investigate differentiation directly, these observations suggest a shift favoring differentiation in CECs treated with TNFα. In line with this, perturbations in glycolysis and ROS generation, both of which we observed in CECs following the TNFα treatment, reportedly lead to increased enteroid differentiation and Lgr5 expression [68]. A separate study reported that TNFR1 knockout mice displayed significantly reduced Lgr5 expression; however, within this same study, colonoids treated with anti-TNFR1 antibody or derived from TNFR1-knockout mice displayed higher Lgr5 expression [69]. Although the latter may be a compensatory effect in the global knockout, it is likely that the length and duration of TNFα treatment may have a differential effect on stem cell expansion in different cell types. 

Along with the reductions in glycolytic acidification observed following butyrate and TNFα co-treatment, our gene expression data provide further evidence for impaired glucose utilization in colonocytes exposed to TNFα and butyrate. Butyrate is the preferred energy source for differentiated CECs [70,71], and germ-free rodents display an energy-deprived colonic phenotype that can be rescued by butyrate [72]; thus, butyrate is necessary for CEC homeostasis. Indeed, butyrate has been shown to alter metabolism in CaCO_2_ cells by hyperacetylating histones in the promoter region of *Pdk4* [73], increasing lipid production from glutamine via a PDK4-dependent mechanism; here, we observed that *Pdk4* transcripts were upregulated when cells were exposed to both butyrate and TNFα. Although we did not measure enzyme activity, PDK4 has been shown to switch the cells away from glycolysis toward fatty acid metabolism [74]. Our data suggest that the presence of butyrate and TNFα may favor lipid metabolism rather than glycolysis. On the other hand, TNFα alone can reduce the expression of the monocarboxylate transporters *slc16a1* and *slc16a3*, fatty acid transporters that can shuttle lactate, pyruvate, and SCFAs such as butyrate [75,76]. In this study, the relative gene expression levels of Slc16a1 were also downregulated with TNFα alone, thus possibly limiting butyrate entry into the cell, but butyrate was able to counteract this. The apparently contradictory inhibition of Slc16a1 expression by high doses of butyrate suggests a negative feedback mechanism of butyrate transport via Slc16a1, as butyrate may regulate its own transport into the cell. Our group has previously shown that the expression of Slc5a8, another luminal butyrate transporter in epithelial cells in the colon, was downregulated in the SHR, which limited the transport of butyrate across the gut epithelium [9]. The downregulation of this transporter may also be correlated with high levels of butyrate that accumulate in the colon of SHRs [9,77], suggesting that the self-regulation of fatty acid transport may be affected both by high butyrate levels and inflammation. However, we were unable to reliably measure this transcript in our cell cultures, and the roles of TNFα and butyrate in the modulation of this transporter remain to be determined. The relative expression levels of Slc16a1 and Slc16a3 were also decreased with TNFα, but not with the co-treatment of butyrate and TNFα, further suggesting a shift away from glycolytic respiration in the presence of TNFα [78]. Further studies should elucidate these effects in vivo and in the context of HTN.

### 4.5. TNFα Increases Biopterin Levels in the Presence and Absence of Butyrate 

Our metabolomics data show an increase in biopterin levels following TNFα treatment. Furthermore, butyrate appeared to augment biopterin levels when co-administered with TNFα, despite not significantly increasing biopterin levels when administered alone. Biopterins are enzymatic co-factors essential for the synthesis of amines such as norepinephrine, serotonin, and histamine, as well as trace amines [79,80], utilized by many vertebrates, invertebrates, bacteria, and fungi [81]. Considering that amines are produced by both the gut epithelium and gut microbiota [82], the possibility of host–microbiota cross-talk is high. Inflammatory cytokines have been reported to increase the de novo synthesis of biopterins in several cell types [83,84]. In the gut, the production of serotonin, for example, can have differential effects, one of which can lead to elevated GI inflammation and the modulation of gut bacteria [85]. Thus, TNFα in the colon may act to further exacerbate inflammation. In HTN, however, it has been shown that the oxidation of biopterin by ROS can reportedly result in biopterin deficiency [86]. Altered biopterin pathway signaling has been reported in the SHR [87,88], while tetrahydrobiopterin administration has been shown to reduce blood pressure in hypertensive rodents and patients [88,89,90]. Considering that TNFα modulated ROS and biopterin production in CECs, the net effect may be diminished in an isolated cell culture and should be further investigated in vivo. 

Of note, a transcriptomics dataset of the colonic epithelium of 3-week-old SHR and WKY rodents [34] shows that the pterin metabolic pathway is differentially expressed in WKY rodents and SHRs [34,35], suggesting broader dysregulation of the pterin metabolic pathway. Within the pterin family, folate metabolism is known to play a significant role in the recycling of fully reduced biopterin [91] and has also been linked to oxidative stress, elevated blood pressure, and insulin resistance in SHRs [92]. Of note, many genes in the biopterin and folate biosynthesis pathway are also differentially expressed in the SHR colonic epithelium [34,35], including GTP cyclohydrolase I (GCH1), aminomethyltransferase (AMT), dihydrofolate reductase (DHFR), methylenetetrahydrofolate dehydrogenase 1 and 1-like (MTHFD1 and MTHFD1L), methylenetetrahydrofolate reductase (MTHFR), serine hydroxymethyltransferases 1 and 2 (SHMT1 and SHMT 2), and solute carrier family 19 member 1 (SLC19a1). MTHFR, the rate-limiting enzyme of folate production, is expressed at higher levels in the SHR colonic epithelium.

Metabolically, biopterins can exert direct effects on isolated mitochondria, acting as an electron transfer cofactor, which, at high concentrations, can inhibit ATP formation by reducing cytochrome oxidase [93]. In line with this, biopterin supplementation has been shown to exert an effect on oxidative respiration in brown adipose tissue [94]. Biopterin can also regulate mitochondrial biogenesis and adenosine monophosphate kinase (AMPK) activity, and conversely, AMPK activity may exert beneficial effects on biopterin levels [86]; this suggests that biopterin plays a role in nutrient sensing, and therefore, may represent a therapeutic target for HTN and metabolic syndrome. 

Butyrate, however, appears to be a relatively weak inducer of biopterin, despite its reported role in inducing the production of serotonin, for example, in the gut [95]; in the present work, butyrate did not increase biopterin levels when applied alone. Thus, in certain circumstances, such as those observed in HTN where TNFα and oxidative stress are elevated, increased biopterin appears to represent a compensatory mechanism, resulting in a metabolic shift that influences mitochondrial bioenergetics and redox cycling to restore intestinal homeostasis. This appears to be largely independent of butyrate, except when butyrate levels are elevated such as during obesity [96]. Future studies are needed to elucidate these potential pathways in the gut in vivo.

## 5. Conclusions

Several inflammatory conditions of the colon involving elevated TNFα present with increased ROS production, microbial dysbiosis, and altered circulating butyrate and other SCFA levels or transport. Butyrate regulates the host–microbiota axis, with important implications for inflammation [80], and the SHR presents with decreased circulating butyrate levels, as well as impaired colonic absorption of butyrate and reduced levels of the high-affinity butyrate transporter, SLC5a8 [9]. The present work addresses an important knowledge gap regarding the interaction between butyrate and TNFα in the colon and suggests that butyrate exerts key effects on TNFα signaling via the mitochondria. Future studies should focus on the interaction between butyrate and TNFα in the context of gut homeostasis as it pertains to calcium signaling, HDAC activity, and the relative contribution of SCFA receptors. As TNFα is known to inhibit the expression of SLC5a8 and is elevated in the colons of SHRs [34], an understanding of the interaction between butyrate and TNFα in the colon could help elucidate the pathophysiology, as well as develop novel therapeutics for inflammatory conditions, of the colon.

## Figures and Tables

**Figure 1 biomolecules-13-00258-f001:**
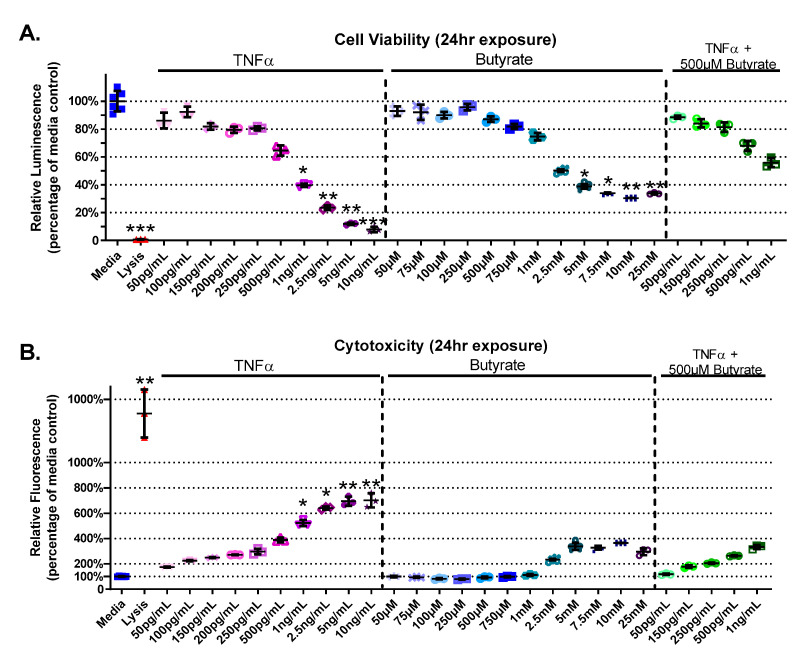
Viability and cytotoxicity after 24 h exposure to butyrate and TNFα. Lysis solution provided with the CellTox Green kit was used as positive control. (**A**) Viability: relative luminescence is graphed for each experimental group and is normalized to the medium control (horizonal bar represents mean relative fluorescence ± standard deviation). (**B**) Cytotoxicity: relative fluorescence is graphed for each experimental group and is normalized to the lysis control (horizonal bar represents mean relative fluorescence ± standard deviation). Asterisks indicate degree of significance from medium control (Kruskal–Wallis test followed by a Dunn’s multiple comparison test relative to the medium control, n = 3/experiment, * = *p* < 0.05, ** = *p* < 0.01, *** = *p* < 0.001). Note: for statistical comparisons made amongst all groups, see Appendix A.

**Figure 2 biomolecules-13-00258-f002:**
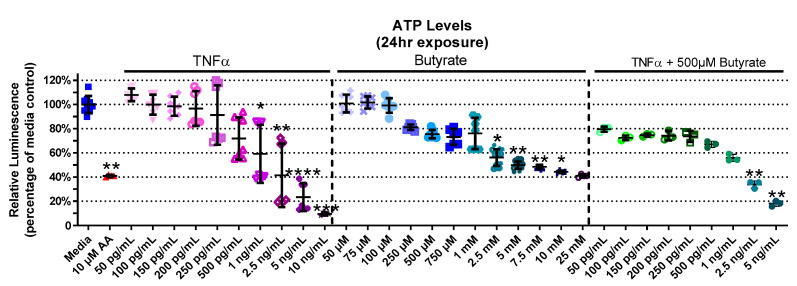
Relative ATP Levels after exposure to butyrate and TNFα for 24 h. Antimycin (AA) was used as a positive control. Relative luminescence is graphed for each experimental group and is normalized to the medium control (horizonal bar represents mean relative fluorescence ± standard deviation). Asterisks indicate degree of significance from medium control (Kruskal–Wallis test followed by a Dunn’s multiple comparison test relative to the medium control, n = 3–4/experiment; two independent experiments were combined; * = *p* < 0.05, ** = *p* < 0.01, *** = *p* < 0.001, **** = *p* < 0.0001). Note: for statistical comparisons made amongst all groups, see Appendix A.

**Figure 3 biomolecules-13-00258-f003:**
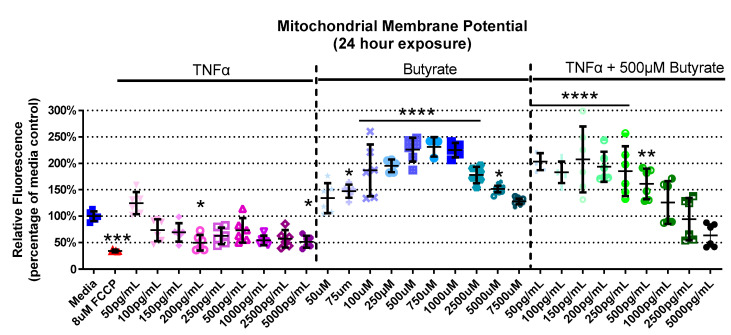
Mitochondrial membrane potential after exposure to butyrate and TNFα at 24 h. FCCP was used as positive control. Relative red/green fluorescence ratio is graphed for each experimental group and is normalized to the medium control (mean relative fluorescence ± standard deviation). Asterisks indicate degree of significance from medium control (Kruskal–Wallis test followed by a Dunn’s multiple comparison test relative to the medium control, n = 3/experiment; significance determined at * = *p* < 0.05, ** = *p* < 0.01, *** = *p* < 0.001, **** = *p* < 0.0001). Note: for statistical comparisons made amongst all groups, see Appendix A.

**Figure 4 biomolecules-13-00258-f004:**
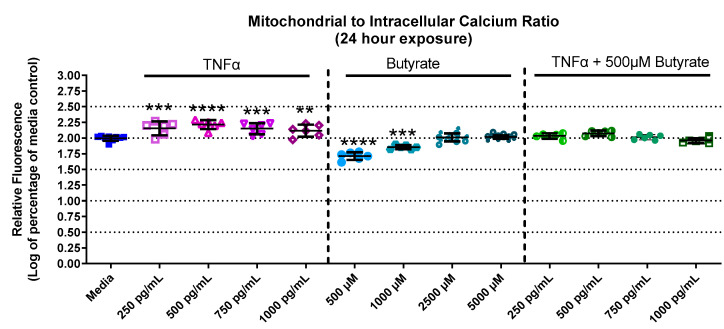
Mitochondrial and intracellular calcium after exposure to butyrate and TNFα at 24 h. Ratio of mitochondrial to intracellular calcium. Fluorescence intensity is graphed for each experimental group and is normalized to the medium control (mean relative fluorescence ± standard deviation). Asterisks indicate degree of significance from medium control (One-way ANOVA followed by a Dunnett’s multiple comparison test, n = 6/experiment; significance determined at ** = *p* < 0.01, *** = *p* < 0.001, **** = *p* < 0.0001).

**Figure 5 biomolecules-13-00258-f005:**
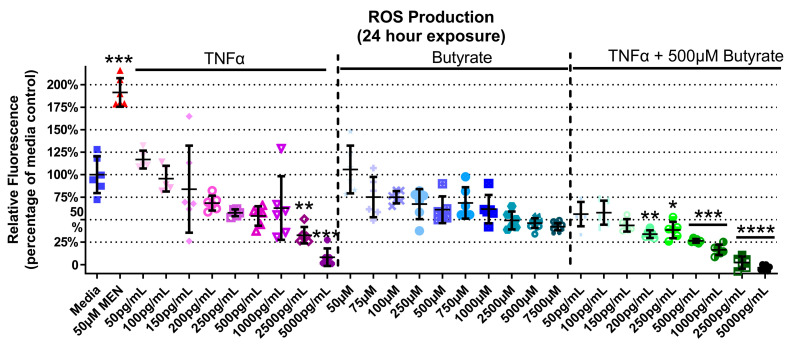
Reactive oxygen species (ROS) production of primary rat colon epithelial cells after exposure to butyrate and TNFα at 24 h. Menadione (MEN) was used as a positive control as a potent inducer of ROS (mean fluorescence ± standard deviation) (n = 3). Asterisks indicate degree of significance from medium control (Kruskal–Wallis test followed by a Dunn’s multiple comparison test relative to the medium control, n = 3; significance determined at * = *p* < 0.05, ** = *p* < 0.01, *** = *p* < 0.001, **** = *p* < 0.0001). Note: for statistical comparisons made amongst all groups, see Appendix A.

**Figure 6 biomolecules-13-00258-f006:**
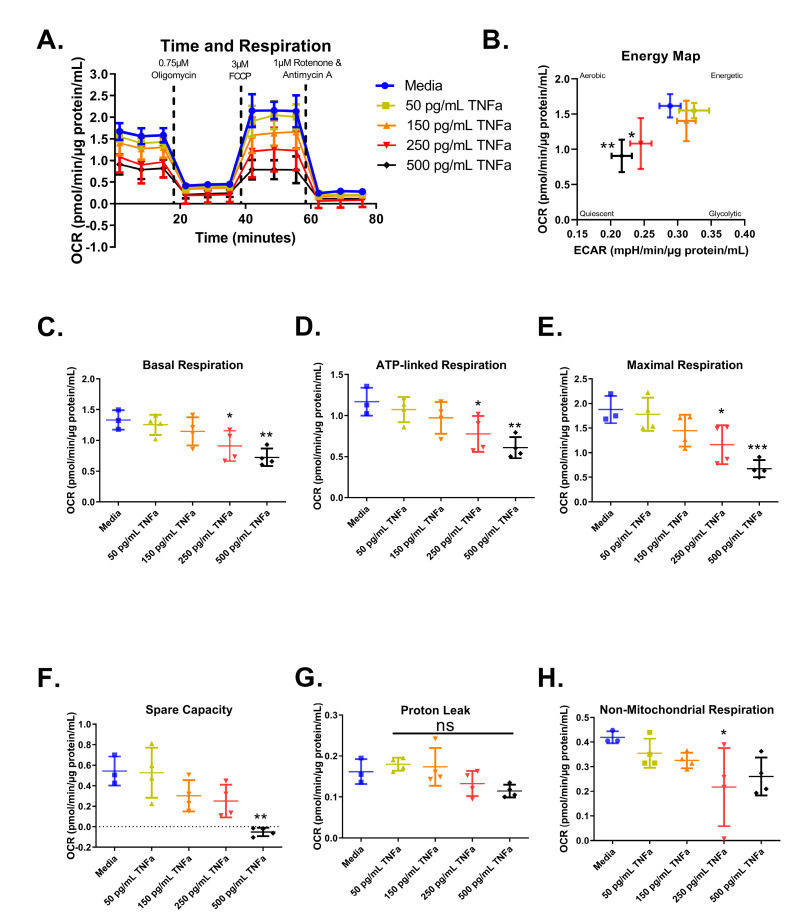
Normalized oxygen consumption rate for primary rat colon epithelial cells after 24 h exposure to TNFα. (**A**) Oxygen consumption rates over time, (**B**) energy map, (**C**) basal respiration, (**D**) oligomycin-induced ATP-linked respiration, (**E**) FCCP-induced maximal respiration, (**F**) spare capacity, (**G**) proton leak, and (**H**) non-mitochondrial respiration. Data are presented as mean ± standard deviation. Asterisks indicate degree of significance from medium control (One-way ANOVA followed by a Dunnett’s multiple comparison test, n = 3 or 4; significance determined at * = *p* < 0.05, ** = *p* < 0.01, *** = *p* < 0.001). NS = not significant.

**Figure 7 biomolecules-13-00258-f007:**
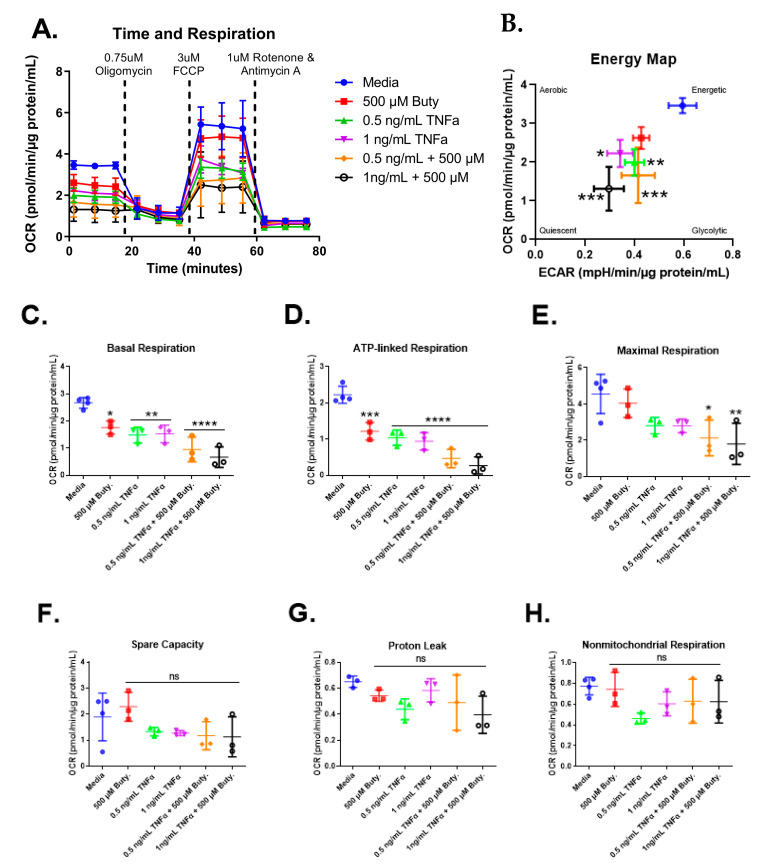
Normalized oxygen consumption rate for primary rat colon epithelial cells after 24 h exposure to butyrate and TNFα. (**A**) Oxygen consumption rates over time, (**B**) energy map, (**C**) basal respiration, (**D**) oligomycin-induced ATP-linked respiration, (**E**) FCCP-induced maximal respiration, (**F**) spare capacity, (**G**) proton leak, and (**H**) non-mitochondrial respiration. Data are presented as mean ± standard deviation. Asterisks indicate degree of significance from medium control (One-way ANOVA followed by a Dunnett’s multiple comparison test, n = 3 or 4; significance determined at * = *p* < 0.05, ** = *p* < 0.01, *** = *p* < 0.001, **** = *p* < 0.0001).

**Figure 8 biomolecules-13-00258-f008:**
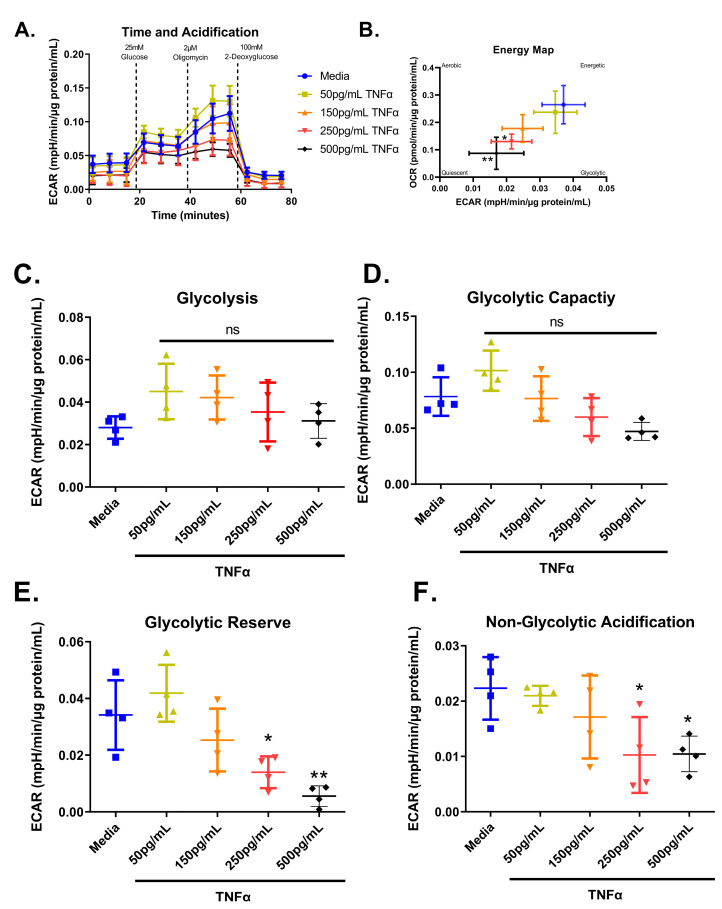
Normalized extracellular acidification rate for primary rat colon epithelial cells after 24-h exposure to TNFα. (**A**) Acidification rates over time, (**B**) energy map, (**C**) glycolysis, (**D**) glycolytic capacity, (**E**) glycolytic reserve, and (**F**) non-glycolytic acidification. Data are presented as mean ± standard deviation. Asterisks indicate degree of significance from medium control (One-way ANOVA followed by a Dunnett’s multiple comparison test, n = 3 or 4; significance determined at * = *p* < 0.05, ** = *p* < 0.01).

**Figure 9 biomolecules-13-00258-f009:**
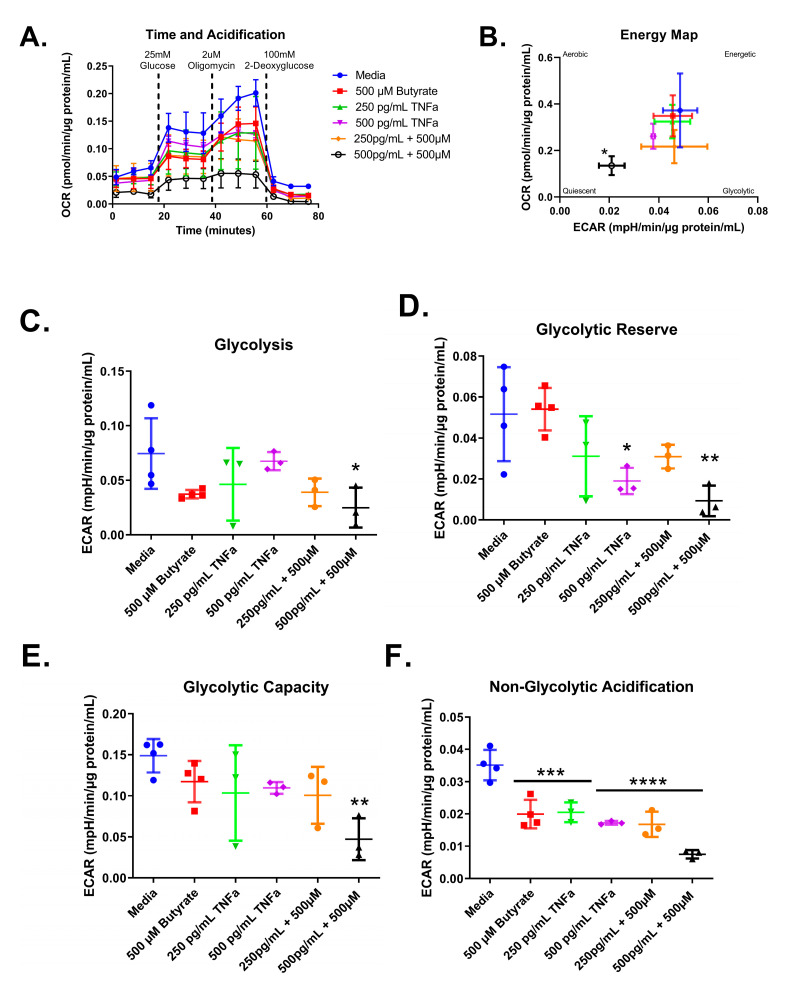
Normalized extracellular acidification rate for primary rat colon epithelial cells after 24 h exposure to butyrate and TNFα. (**A**) Acidification rates over time, (**B**) energy map, (**C**) glycolysis, (**D**) glycolytic capacity, (**E**) glycolytic reserve, and (**F**) non-glycolytic acidification. Data are presented as mean ± standard deviation. Asterisks indicate degree of significance from medium control (One-way ANOVA followed by a Dunnett’s multiple comparison test, n = 3 or 4; significance determined at * = *p* < 0.05, ** = *p* < 0.01, *** = *p* < 0.001, **** = *p* < 0.0001).

**Figure 10 biomolecules-13-00258-f010:**
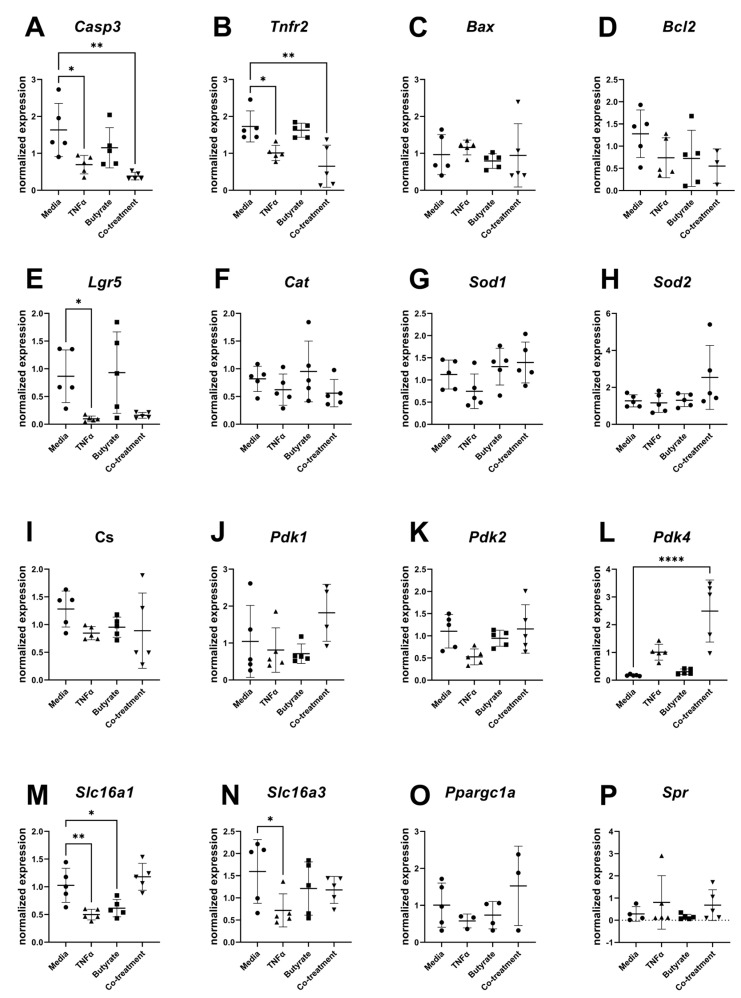
Relative gene expression for primary rat colon epithelial cells after 24 h exposure to butyrate and TNFα: (**A**) Casp3, (**B**) Tnfr2, (**C**) Bax, (**D**) Bcl2, (**E**) Lgr5, (**F**) Cat, (**G**) Sod1, (**H**) Sod2, (**I**) Cs, (**J**) Pdk1, (**K**) Pdk2, (**L**) Pdk4, (**M**) Slc16a1, (**N**) Slc16a3, (**O**) Ppargc1a, and (**P**) Spr. Data are presented as mean ± standard deviation. Asterisks indicate degree of significance from medium control (One-way ANOVA followed by a Dunnett’s multiple comparison test, n = 5; significance determined at * = *p* < 0.05, ** = *p* < 0.01, **** = *p* < 0.0001).

**Figure 11 biomolecules-13-00258-f011:**
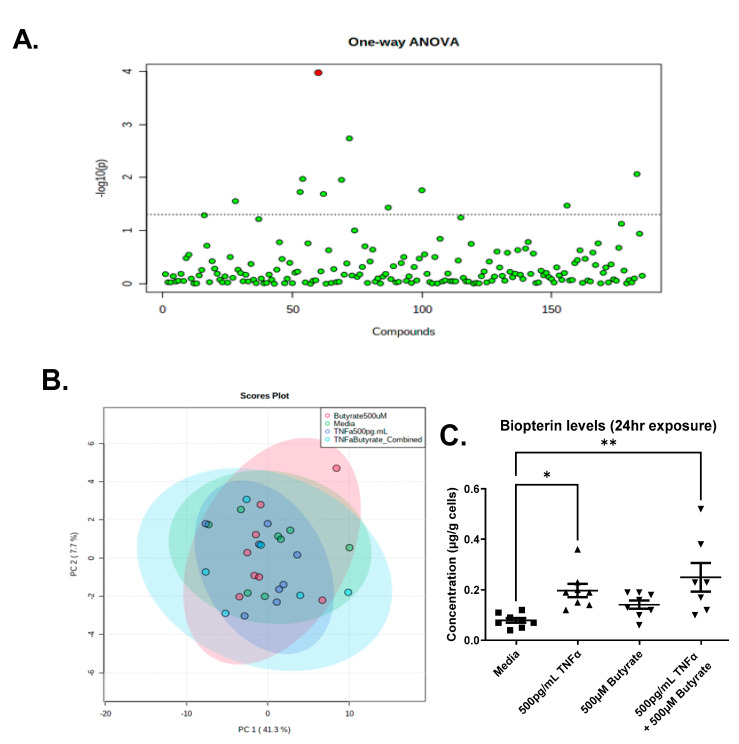
Metabolomics analyses reveal elevated biopterin in rodent CECs following treatment with TNFα. (**A**) One-Way ANOVA showing Biopterin as the only differentially abundant metabolite following FDR correction; (**B**) PCA Scores plot for all treatments; and (**C**) Semi-quantitative levels of biopterin in primary rat colon epithelial cells after 24 h exposure to butyrate and TNFα. Data are presented as mean ± standard deviation. Asterisks indicate degree of significance from medium control (One-way ANOVA followed by a Dunnett’s multiple comparison test, n = 7–8; significance determined at * = *p* < 0.05, ** = *p* < 0.01).

**Figure 12 biomolecules-13-00258-f012:**
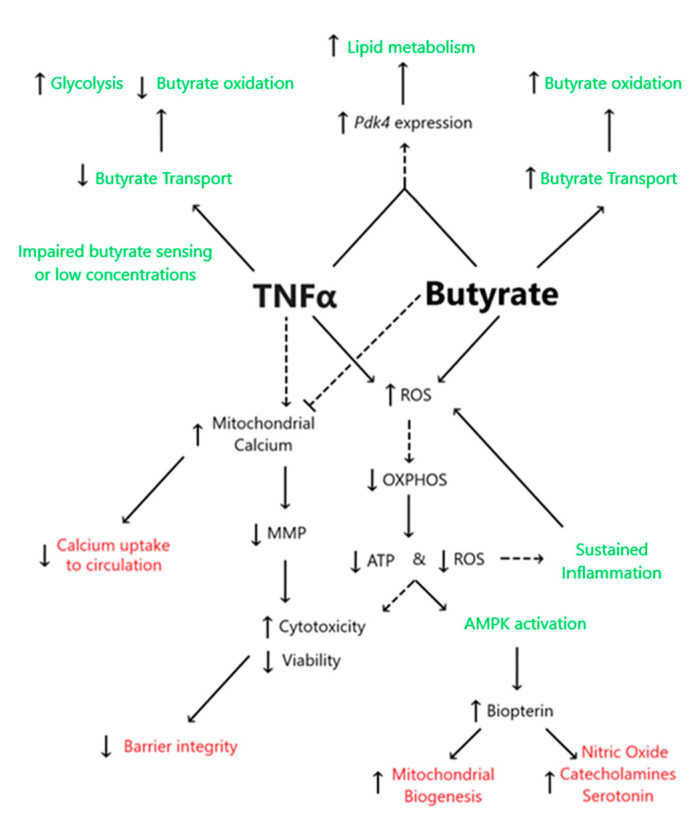
Proposed interactions between butyrate and TNFα in rodent CECs. Dashed lines represent predicted associations, whereas solid lines represent associations supported in the literature. Black text highlights endpoints measured within the present work, green text represents non-measured endpoints obtained from the literature, and red text represents information regarding butyrate and TNFα interactions.

## Data Availability

Not applicable.

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
