# Peer review of "Interaction between Butyrate and Tumor Necrosis Factor α in Primary Rat Colonocytes"

_biomolecules, 2023, doi:10.3390/biom13020258_

Round 1
Reviewer 1 Report
The paper by Souders II et al. investigated if butyrate modifies TNFα effedts in primary rodent colonic cells. The use of rodent cells (and not human) should be explained. The statistical analysis is rather arbitrary, since three different post hoc tests ((Dunnet’s, Tukey’s and LSD) are arbitrarily applied (with no clear reason). The statistical differences among the groups are not clearly demonstrated, and authors often rely on percentages (and “more significant” p values) to arbitrarily explain the difference among groups (not considering the absence of difference in the post hoc test). The paper lacks didactics, and, considering the metabolomics results, suffers from confirmation bias.
· Introduction: The reviewer understands that the introduction covers the established relationship between butyrate and TNF-α in hypertension, but the importance of TNF-α in Inflammatory bowel disease (IBD, Crohn's disease and ulcerative colitis), in which is neglected is this section.
· Introduction: The summary of methodologies (lines 85-90) is not necessary in this section. It makes the introduction lengthy.
· Methods: Why not using a human cell line? Authors should provide a proper reason for this methodological choice.
· Methods: the number of biological and technical replicates for each experiment should be clearly stated in the methods section.
· Methods: Three different post hoc tests are arbitrarily used (Dunnet’s, Tukey’s and LSD). Authors should provide a proper reason for this.
· Results: Authors do not present the exact “p” values for each comparison, only the threshold (P<0.05). Please provide the exact p values for each test.
· Figures 1 to 5: “Asterisks indicate degree of significance from media control”. It is not clear what the authors meant by that. It is rather arbitrary.
· Figures 1 to 5: Considering that authors only considered statistical differences compared to control (according to the captions of the figures), “Asterisks indicate degree of significance from media control”, it is difficult to point (only looking at the data or only considering percentages) if the cotreatment was different from TNF-a or Butyrate. These results lack scientific rigor, and authors should consider the other comparisons.
· Results description: In general, the description of results is also lengthy and full of methodological (statistical) details. Let me give you an example (lines 331-333): “Experimental groups varied for cell viability F(28, 61) = 256.7, P<0.0001) and cytotoxicity 331 (F(28, 61) = 157.8, P<0.0001) in primary rat CECs with butyrate and TNFα concentration at 24 332 hours (Figure 1A-B). Following a Dunnett’s post-hoc correction”. These details are not necessary in the results. Authors should rethink their result description strategy in order to sound more didactic.
· Metabolomics: PCA analysis shows a great similarity between treatments. This should be clearly stated by the authors. Of 185 metabolites, one (biopterin) showed significant showed significant results compared to control only. In fact, cotreatment did not promoted TNF-a-induced increased in biopterin, it was only “more significant” (it is not clear what does this mean). The interpretation of the results is rather arbitrary, and the authors “hide” this lack of difference between TNF-a and cotreatment using the asterisks. In other results (e.g., Fig 7), when authors see a statistical difference among these groups, they use letters to describe them. Authors should only consider statistical difference among groups (indicated by post hoc tests).
Author Response
Reviewer 1
The paper by Souders II et al. investigated if butyrate modifies TNFα effects in primary rodent colonic cells. The use of rodent cells (and not human) should be explained. The statistical analysis is rather arbitrary, since three different post hoc tests ((Dunnet’s, Tukey’s and LSD) are arbitrarily applied (with no clear reason). The statistical differences among the groups are not clearly demonstrated, and authors often rely on percentages (and “more significant” p values) to arbitrarily explain the difference among groups (not considering the absence of difference in the post hoc test). The paper lacks didactics, and, considering the metabolomics results, suffers from confirmation bias.
Response: Thank you for your valuable comments, we address the reviewer’s concerns below and throughout the manuscript.
- Introduction: The reviewer understands that the introduction covers the established relationship between butyrate and TNF-α in hypertension, but the importance of TNF-α in Inflammatory bowel disease (IBD, Crohn's disease and ulcerative colitis), in which is neglected is this section.
Response: Thank you, we have added this to the introduction along with references. We kept this brief as to not detract from the focus of the study.
- Introduction: The summary of methodologies (lines 85-90) is not necessary in this section. It makes the introduction lengthy.
Response: We have removed this from the Introduction.
- Methods: Why not using a human cell line? Authors should provide a proper reason for this methodological choice.
Response: Our group works in rodent models for hypertension, and we use the rodent colonocytes to elucidate chemicals in species-specific cells. Our aim is to explore such mechanisms in spontaneously hypertensive rats in future studies before we move onto effects in humans. We agree with the reviewer’s suggestion and future work in human cells will strengthen our understanding of the mechanisms to provide translational context. We added a sentence in the manuscript to explain our model.
- Methods: the number of biological and technical replicates for each experiment should be clearly stated in the methods section.
Response: These have been added to each section. These are also reported in figure captions now.
- Methods: Three different post hoc tests are arbitrarily used (Dunnet’s, Tukey’s and LSD). Authors should provide a proper reason for this.
Response: We often go with the recommendations of the statistical software package – PRISM. The LSD is more typical of metabolomics data, and this was the preferred approach for this analysis based on MetaboAnalyst. Depending on whether the test conducted was Kruskal-Wallis or ANOVA, the appropriate post-hoc test was used. This is now clearly indicated throughout the manuscript where appropriate.
- Results: Authors do not present the exact “p” values for each comparison, only the threshold (P<0.05). Please provide the exact p values for each test. It is not clear what the authors meant by that. It is rather arbitrary.
Response: For some graphs, we give the threshold only because there are many doses that are included in the group. For example, “Butyrate at 5 mM and above reduced viability compared to media alone (P<0.05).”, referring to multiple concentrations. To be more complete, we include statistics output for each test in the Supplemental Data Table 3.
- Figures 1 to 5: Considering that authors only considered statistical differences compared to control (according to the captions of the figures), “Asterisks indicate degree of significance from media control”, it is difficult to point (only looking at the data or only considering percentages) if the cotreatment was different from TNF-a or Butyrate. These results lack scientific rigor, and authors should consider the other comparisons.
Response: To simplify the presentation of the graphs, we show comparisons only to the media control. With numerous concentrations and treatments, there are more than 300+ comparisons that are made for each graph which is not possible to represent on a figure, but we include full intergroup statistical comparisons in the Supplemental Data Table 3.
- Results description: In general, the description of results is also lengthy and full of methodological (statistical) details. Let me give you an example (lines 331-333): “Experimental groups varied for cell viability F(28, 61) = 256.7, P<0.0001) and cytotoxicity 331 (F(28, 61) = 157.8, P<0.0001) in primary rat CECs with butyrate and TNFα concentration at 24 332 hours (Figure 1A-B). Following a Dunnett’s post-hoc correction”. These details are not necessary in the results. Authors should rethink their result description strategy in order to sound more didactic.
Response: We have reduced the wordiness of the results, taking care to keep the description of the stats that are relevant to our overall conclusions. The results sections have now also been written in a more didactic format.
- Metabolomics:PCA analysis shows a great similarity between treatments. This should be clearly stated by the authors. Of 185 metabolites, one (biopterin) showed significant showed significant results compared to control only. In fact, cotreatment did not promoted TNF-a-induced increased in biopterin, it was only “more significant” (it is not clear what does this mean). The interpretation of the results is rather arbitrary, and the authors “hide” this lack of difference between TNF-a and cotreatment using the asterisks. In other results (e.g., Fig 7), when authors see a statistical difference among these groups, they use letters to describe them. Authors should only consider statistical difference among groups (indicated by post hoc tests).
Response: This has now been clearly stated and interpretation of all results has been adjusted to strictly follow the results of the significance tests.
Reviewer 2 Report
This study aimed at investigating to which extent butyrate can counteract the negative effects of TNF-alpha on cytotoxicity, inflammation, oxidative stress in primary rat colonocytes. The article is very interesting, the amount of work is impressive. I have only few comments to the authors :
Line 6-15 : there is a problem with the number in the affiliations
Line 72 : via ‘the’ increase in glycolysis
Introduction/conclusion : is there any nutritional strategy based on butyrate supplementation to prevent hypertension in humans ?
Line 97 : Is it sodium or calcium butyrate ?
Line 308 : did you also test homoscedasticity before running the ANOVA ? I have the feeling that non-parametric tests might be more appropriate on some specific parameters
Material and methods : What are the physiological concentration of tnf-alpha and butyrate ? How do they compare with the concentrations tested in this work ?
Figure 4 : repaste the figure, missing stars on 5 µM FCCP I guess
Figure 5 : missing stars on 50 µM MEN group I guess
Figure 7 C-H : why making 2 by 2 group comparison here and not in the other
Figure 7, 9 : repeat TNF-a and Buty in the legend of the orange and black groups
Figure 11. B : this figure is not easy to read
Author Response
Reviewer 2
Comments and Suggestions for Authors
This study aimed at investigating to which extent butyrate can counteract the negative effects of TNF-alpha on cytotoxicity, inflammation, oxidative stress in primary rat colonocytes. The article is very interesting, the amount of work is impressive. I have only few comments to the authors:
Response: Thank you very much for the positive response.
Line 6-15 : there is a problem with the number in the affiliations
Response: These affiliations have now been corrected for all the authors. Thank you for catching this.
Line 72 : via ‘the’ increase in glycolysis
Response: This edit has been made.
Introduction/conclusion: is there any nutritional strategy based on butyrate supplementation to prevent hypertension in humans?
Response: There have been no studies, to our knowledge, that showed effects of dietary butyrate supplementation on blood pressure in humans or rodents. However, low circulating butyrate has been associated with hypertension, and supplementation with butyrate s.c. (to bypass the GI tract) can decrease blood pressure in some forms of experimental hypertension.
Line 97: Is it sodium or calcium butyrate?
Response: Sodium, and this has now been clarified in the Methods.
Line 308: did you also test homoscedasticity before running the ANOVA? I have the feeling that non-parametric tests might be more appropriate on some specific parameters.
Response: Thank you, we have performed a Kruskal-Wallis test, followed by a Dunn’s post hoc test for those data that did not meet requirements of an ANOVA. There were some groups that did not conform to normality following a log transformation. These we analyze all the data using KS test. This has now been clarified in the text – as to which endpoints were analyzed by non-parametric versus parametric tests.
Material and methods: What are the physiological concentration of tnf-alpha and butyrate? How do they compare with the concentrations tested in this work?
Response: We have now added a new paragraph to address this. We add “”
Figure 4: repaste the figure, missing stars on 5 µM FCCP I guess
Response: Correct, thank you – these have now been added.
Figure 5: missing stars on 50 µM MEN group I guess
Response: These have now been added, thank you.
Figure 7 C-H: why making 2 by 2 group comparison here and not in the other
Response: These endpoints are derived from Panel A. Panel A consists of the entire assay and statistical evaluation is not conducted until individual endpoints (basal respiration, maximum respiration, etc) are calculated.
Figure 7, 9: repeat TNF-a and Buty in the legend of the orange and black groups
Response: This has been repeated.
Figure 11. B : this figure is not easy to read
Response: We have increased the size of the figure.
Reviewer 3 Report
The manuscript “Interaction between Butyrate and Tumor Necrosis Factor α in 2 Primary Rat Colonocytes” discussed the influence of butyrate and TNFα on colonocytes inflammatory, it is very interesting, however, there are also some problems needing to be solved before it can be accepted.
(1) The authors mentioned that co-treatment of 500 µ M butyrate offered some protection, as a result, in figure 2 -4, the authors showed the effect of the co-treatment of 500 µ M butyrate with different doses of TNFα, however, the authors also stated the results of the co-treatment of 250 µ M butyrate with TNFα at 72 hours, however, without detailed figures or data, the authors should explain the reasons they choose different doses(250 µ M,500µ M butyrate), and show figures or tables to present the data at 72 hours.
(2) In this paper, the author used a range of doses for TNFα (50 pg/mL to 10 ng/mL), the authors should explain the reasons to choose these doses, which is the normal dose in physiological condition, which represent the dose in pathological conditions?
(3)In Figure 11B , there is no significant difference between these groups, the author should explain the PCA data clearly.
Author Response
Reviewer 3
The manuscript “Interaction between Butyrate and Tumor Necrosis Factor α in 2 Primary Rat Colonocytes” discussed the influence of butyrate and TNFα on colonocytes inflammatory, it is very interesting, however, there are also some problems needing to be solved before it can be accepted.
Response: We thank the review for the positive comments.
(1) The authors mentioned that co-treatment of 500 µ M butyrate offered some protection, as a result, in figure 2 -4, the authors showed the effect of the co-treatment of 500 µ M butyrate with different doses of TNFα, however, the authors also stated the results of the co-treatment of 250 µ M butyrate with TNFα at 72 hours, however, without detailed figures or data, the authors should explain the reasons they choose different doses(250 µ M,500µ M butyrate), and show figures or tables to present the data at 72 hours.
Response: We agree, and we now include the figures for 4 and 72 hours into the Supplemental files. We also add a section on the rationale for concentrations used and why we focused on 24 hours (response was greatest for several endpoints at this timepoint).
(2) In this paper, the author used a range of doses for TNFα (50 pg/mL to 10 ng/mL), the authors should explain the reasons to choose these doses, which is the normal dose in physiological condition, which represent the dose in pathological conditions?
Response: Reviewer 2 also requested this information. We added a paragraph that clearly describes our rationale for the concentrations tested.
(3) In Figure 11B , there is no significant difference between these groups, the author should explain the PCA data clearly.
Response: We increase the size of the figure and explain the PCA in the text. More information is now been added with regards to explaining these results.
Round 2
Reviewer 1 Report
The authors adressed most of the reviewer's suggestions. The manuscript should be published.
Reviewer 3 Report
The authors have addressed all the mentioned questions.